# PREDICTING CLASSIFICATION ACCURACY WHEN ADDING NEW UNOBSERVED CLASSES

**Yuli Slavutsky, Yuval Benjamini**
Department of Statistics and Data Science
The Hebrew University of Jerusalem
Jerusalem, Israel
{yuli.slavutsky, yuval.benjamini}@mail.huji.ac.il

## ABSTRACT

Multiclass classifiers are often designed and evaluated only on a sample from the classes on which they will eventually be applied. Hence, their final accuracy remains unknown. In this work we study how a classifier's performance over the initial class sample can be used to extrapolate its expected accuracy on a larger, unobserved set of classes. For this, we define a measure of separation between correct and incorrect classes that is independent of the number of classes: the *reversed ROC* (rROC), which is obtained by replacing the roles of classes and data-points in the common ROC. We show that the classification accuracy is a function of the rROC in multiclass classifiers, for which the learned representation of data from the initial class sample remains unchanged when new classes are added. Using these results we formulate a robust neural-network-based algorithm, *CleaneX*, which learns to estimate the accuracy of such classifiers on arbitrarily large sets of classes. Unlike previous methods, our method uses both the observed accuracies of the classifier and densities of classification scores, and therefore achieves remarkably better predictions than current state-of-the-art methods on both simulations and real datasets of object detection, face recognition, and brain decoding.

## 1 INTRODUCTION

Advances in machine learning and representation learning led to automatic systems that can identify an individual class from very large candidate sets. Examples are abundant in visual object recognition (Russakovsky et al., 2015; Simonyan & Zisserman, 2014), face identification (Liu et al., 2017b), and brain-machine interfaces (Naselaris et al., 2011; Seeliger et al., 2018). In all of these domains, the possible set of classes is much larger than those observed at training or testing.

Acquiring and curating data is often the most expensive component in developing new recognition systems. A practitioner would prefer knowing early in the modeling process whether the data-collection apparatus and the classification algorithm are expected to meet the required accuracy levels. In large multi-class problems, the pilot data may contain considerably fewer classes than would be found when the system is deployed (consider, for example, the case in which researchers develop a face recognition system that is planned to be used on 10,000 people, but can only collect 1,000 in the initial development phase). This increase in the number of classes changes the difficulty of the classification problem and therefore the expected accuracy. The magnitude of change varies depending on the classification algorithm and the interactions between the classes: usually classification accuracy will deteriorate as the number of classes increases, but this deterioration varies across classifiers and data-distributions. For pilot experiments to work, theory and algorithms are needed to estimate how accuracy of multi-class classifiers is expected to change when the number of classes grows. In this work, we develop a prediction algorithm that observes the classification results for a small set of classes, and predicts the accuracy on larger class sets.

In large multiclass classification tasks, a representation is often learned on a set of $k_1$ classes, whereas the classifier is eventually used on a new larger class set. On the larger set, classification

can be performed by applying simple procedures such as measuring the distances in an embedding space between a new example $x \in \mathcal{X}$ and labeled examples associated with the classes $y_i \in \mathcal{Y}$. Such classifiers, where the score assigned to a data point $x$ to belong to a class $y$ is independent of the other classes, are defined as *marginal classifiers* (Zheng et al., 2018). Their performance on the larger set describes how robust the learned representation is. Examples of classifiers that are marginal when used on a larger class set include siamese neural networks (Koch et al., 2015), one-shot learning (Fei-Fei et al., 2006) and approaches that directly optimize the embedding (Schroff et al., 2015). Our goal in this work is to estimate how well a given marginal classifier will perform on a large unobserved set of $k_2$ classes, based on its performance on a smaller set of $k_1$ classes.

Recent works (Zheng & Benjamini, 2016; Zheng et al., 2018) set a probabilistic model for rigorously studying this problem, assuming that the $k_1$ available classes are sampled from the same distribution as the larger set of $k_2$ classes. Following the framework they propose, we assume that the sets of $k_1$ and $k_2$ classes on which the classifier is trained and evaluated are sampled independently from an infinite continuous set $\mathcal{Y}$ according to $Y_i \sim P_Y(y)$, and for each class, $r$ data points are sampled independently from $\mathcal{X}$ according to the conditional distribution $P_{X|Y}(x \mid y)$. In their work, the authors presented two methods for predicting the expected accuracy, one of them originally due to Kay et al. (2008). We cover these methods in Section 2.

As a first contribution of this work (Section 3), we provide a theoretical analysis that connects the accuracy of marginal classifiers to a variant of the receiver operating characteristic (ROC) curve, which is achieved by reversing the roles of classes and data points in the common ROC. We show that the *reversed ROC* (rROC) measures how well a classifier's learned representation separates the correct from the incorrect classes of a given data point. We then prove that the accuracy of marginal classifiers is a function of the rROC, allowing the use of well researched ROC estimation methods (Gonçalves et al., 2014; Bhattacharya & Hughes, 2015) to predict the expected accuracy. Furthermore, the reversed area under the curve (rAUC) equals the expected accuracy of a binary classifier, where the expectation is taken over all randomly selected pairs of classes.

We use our results regarding the rROC to provide our second contribution (Section 4): *CleaneX* (Classification Expected Accuracy Neural EXtrapolation), a new neural-network-based method for predicting the expected accuracy of a given classifier on an arbitrarily large set of classes[1]. *CleaneX* differs from previous methods by using both the raw classification scores and the observed classification accuracies for different class-set sizes to calibrate its predictions. In Section 5 we verify the performance of *CleaneX* on simulations and real data-sets. We find it achieves better overall predictions of the expected accuracy, and very few "large" errors, compared to its competitors. We discuss the implications, and how the method can be used by practitioners, in Section 6.

## 1.1 PRELIMINARIES AND NOTATION

In this work $x$ are data points, $y$ are classes, and when referred to as random variables they are denoted by $X, Y$ respectively. We denote by $y(x)$ the correct class of $x$, and use $y^*$ when $x$ is implicitly understood. Similarly, we denote by $y'$ an incorrect class of $x$.

We assume that for each $x$ and $y$ the classifier $h$ assigns a score $S_y(x)$, such that the predicted class of $x$ is $\arg\max_y S_y(x)$. On a given dataset of $k$ classes, $\{y_1, \ldots, y_k\}$, the accuracy of the trained classifier $h$ is the probability that it assigns the highest score to the correct class

$$\mathcal{A}(y_1, \ldots, y_k) = P_X(S_{y^*}(x) \geq max_{i=1}^k S_{y_i}(x)) \tag{1}$$

where $P_X$ is the distribution of the data points $x$ in the sample of classes. Since $r$ points are sampled from each class, $P_X$ assumes a uniform distribution over the classes within the given sample.

An important quantity for a data point $x$ is the probability of the correct class $y^*$ to outscore a randomly chosen incorrect class $Y' \sim P_{Y|Y \neq y^*}$, that is $C_x = P_{Y'}(S_{y^*}(x) \geq S_{y'}(x))$. This is the cumulative distribution function of the incorrect scores, evaluated at the value of the correct score.

We denote the expected accuracy over all possible subsets of $k$ classes from $\mathcal{Y}$ by $\mathbb{E}_k[\mathcal{A}]$ and its estimator by $\hat{\mathbb{E}}_k[\mathcal{A}]$. We refer to the curve of $\mathbb{E}_k[\mathcal{A}]$ at different values of $k \geq 2$ as the *accuracy curve*. Given a sample of $K$ classes, the average accuracy over all subsets of $k \leq K$ classes from the sample is denoted by $\bar{\mathcal{A}}_k^K$.

---

[1]Code is publicly available at: https://github.com/YuliSl/CleaneX

## 2 RELATED WORK

Learning theory provides bounds of sample complexity in multiclass classification that depend on the number of classes (Shalev-Shwartz & Ben-David, 2014), and the extension to large mutliclass problems is a topic of much interest (Kuznetsov et al., 2014; Lei et al., 2015; Li et al., 2018). However, these bounds cannot be used to estimate the expected accuracy. Generalization to out-of-label accuracy includes the work of Jain & Learned-Miller (2010). The generalization of classifiers from datasets with few classes to larger class sets include those of Oquab et al. (2014) and Griffin et al. (2007), and are closely related to transfer learning (Pan et al., 2010) and extreme classification (Liu et al., 2017a). More specific works include that of Abramovich & Pensky (2019), which provides lower and upper bounds for the distance between classes that is required in order to achieve a given accuracy.

Kay et al. (2008), as adapted by Zheng et al. (2018), propose to estimate the accuracy of a marginal classifier on a given set of $k$ classes by averaging over $x$ the probability that its correct class outscores a single random incorrect class, raised to the power of $k-1$ (the number of incorrect classes in the sample), that is

$$\mathbb{E}_k[\mathcal{A}] = \mathbb{E}_X[P_{Y'}(S_{y^*}(x) \geq S_{y'}(x))^{k-1}] = \mathbb{E}_x[C_x^{k-1}]. \tag{2}$$

Therefore, the expected accuracy can be predicted by estimating the values of $C_x$ on the available data. To do so, the authors propose using kernel density estimation (KDE) choosing the bandwidth with pseudo-likelihood cross-validation (Cao et al., 1994).

Zheng et al. (2018) define a discriminability function

$$D(u) = P_X\left(P_{Y'}\left(S_{y^*}(x) > S_{y'}(x)\right) \leq u\right), \tag{3}$$

and show that for marginal classifiers, the expected accuracy at $k$ classes is given by

$$\mathbb{E}_k[\mathcal{A}] = 1 - (k-1) \int_0^1 D(u) u^{k-2} du. \tag{4}$$

The authors assume a non-parametric regression model with pre-chosen basis functions $b_j$, so that $D(u) = \sum_j \beta_j b_j$. To obtain $\hat{\beta}$ the authors minimize the mean squared error (MSE) between the resulting estimation $\hat{\mathbb{E}}_k[\mathcal{A}]$ and the observed accuracies $\bar{\mathcal{A}}_k^{k_1}$.

## 3 REVERSED ROC

In this section we show that the expected accuracy, $\mathbb{E}_k[\mathcal{A}]$, can be better understood by studying an ROC-like curve. To do so, we first recall the definition of the common ROC: for two classes in a setting where one class is considered as the positive class and the other as the negative one, the ROC is defined as the graph of the true-positive rate (TPR) against the false-positive rate (FPR) (Fawcett, 2006). The common ROC curve represents the separability that a classifier $h$ achieves between data points of the positive class and those of the negative one. At a working point in which the FPR of the classifier is $u$, we have $\text{ROC}(u) = \text{TPR}\left(\text{FPR}^{-1}(u)\right)$.

In a multiclass setting, we can define $\text{ROC}_y$ for each class $y$ by considering $y$ as the positive class, and the union of all other classes as the negative one. An adaptation of the ROC for this setting can be defined as the expectation of $\text{ROC}_y$ over the classes, that is $\overline{\text{ROC}}(u) = \int_y \text{ROC}_y(u) \, dP(y)$. In terms of classification scores, we have $\text{TPR}_y(t) = P_X(S_y(x) > t \mid y(x) = y)$, $\text{FPR}_y(t) = P_X(S_y(x) > t \mid y(x) \neq y)$ and thus $\text{FPR}_y^{-1}(u) = \sup_t \{P_X(S_y(x) > t \mid y(x) \neq y) \geq u\}$.

Here, we single out each time one of the classes $y$ and compare the score of the data points that belong to this class with the score of those that do not. However, when the number of classes is large, we could instead single out a data point $x$ and compare the score that it gets for the correct class with the scores for the incorrect ones. This reverse view is formalized in the following definition, where we exchange the roles of data points $x$ and classes $y$, to obtain the reversed ROC:

**Definition 1.** *Given a data point $x$, its corresponding* reversed true-positive *rate is*

$$rTPR_x(t) = \begin{cases} 1 & S_{y^*}(x) > t \\ 0 & S_{y^*}(x) \leq t \end{cases} \tag{5}$$

*The* reversed false-positive *rate is*

$$rFPR_x(t) = P_{Y'}(S_{y'}(x) > t) \tag{6}$$

*and accordingly*

$$rFPR_x^{-1}(u) = \sup_t \{P_{Y'}(S_{y'}(x) > t) \geq u\}. \tag{7}$$

*Consequently, the* reversed ROC *is*

$$rROC_x(u) = rTPR_x\left(rFPR_y^{-1}(u)\right) = \begin{cases} 1 & S_{y^*}(x) > \sup_t \{P_{Y'}(S_{y'}(x) > t) \geq u\} \\ 0 & otherwise \end{cases} \tag{8}$$

*and the* average reversed ROC *is[2]*

$$\overline{rROC}(u) = \int_{\mathcal{X}} rROC_x(u)\, dP(x). \tag{9}$$

Since $P_{Y'}(S_{y'}(x) > t)$ is a decreasing function of $t$, it can be seen that $rROC_x(u) = 1$ iff $u > P_{Y'}(S_{y'}(x) > S_{y^*}) = 1 - C_x$ (see Proposition 1 in Appendix A). However, even though $rROC_x$ is a step function, the $\overline{rROC}$ resembles a common ROC curve, as illustrated in Figure 1.

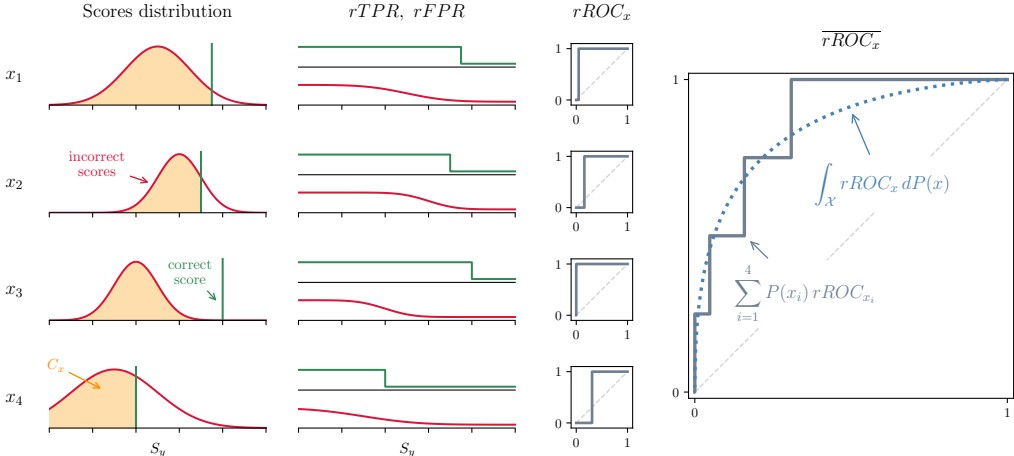

Figure 1: The reversed ROC. The leftmost column shows an example of the score distributions of four data points: the distribution of scores of incorrect classes (red), and the score of the correct class (green). The yellow shaded area is the CDF of the incorrect scores distribution evaluated at the correct score, that is $C_x$. The second column shows the corresponding rTPR (green, top) and rFPR (red, bottom). The third column depicts the resulting $rROC_x$ curves. The rightmost plot presents the average $\overline{rROC}$ over the four data points (solid grey); as the number of averaged data points grows, the $\overline{rROC}$ curve becomes smoother (dotted blue).

### 3.1 THE REVERSED ROC AND CLASSIFICATION ACCURACY

In what follows, we show that the classification accuracy can be expressed using the average reversed ROC curve. We assume a marginal classifier which assigns scores without ties, that is for all $x$ and all $y_i \neq y_j$ we have $S_{y_i}(x) \neq S_{y_j}(x)$ almost surely. In such cases the following theorem holds:

**Theorem 1.** *The expected classification accuracy at $k$ classes is*

$$\mathbb{E}_k[\mathcal{A}] = 1 - (k-1)\int_0^1 \left(1 - \overline{rROC}(1-u)\right) u^{k-2} du. \tag{10}$$

---

[2]Note that $dP(x)$ in Equation equation 9 assumes a uniform distribution with respect to a given sample of classes $\{y_1, \ldots, y_k\}$ and their corresponding data points.

To prove this theorem we show that

$$1 - \overline{\text{rROC}}(1 - u) = P_X \left( P_{Y'} \left( S_{y^*}(x) > S_{y'}(x) \right) \leq u \right) = D(u) \tag{11}$$

and the rest follows immediately from the results of Zheng et al. (2018) (see Equation 4). We provide the detailed proof in Appendix A.

Now, using the properties of the rROC we get

$$\mathbb{E}_k[\mathcal{A}] = 1 - (k - 1) \int_0^1 \left( 1 - \overline{\text{rROC}}(1 - u) \right) u^{k-2} du$$

$$= \int_0^1 \left( \int_{\mathcal{X}} \text{rROC}_x(1 - u) dP(x) \right) (k - 1) u^{k-2} du$$

$$= \int_{\mathcal{X}} \int_0^{C_x} (k - 1) u^{k-2} du \, dP(x) = \int_{\mathcal{X}} C_x^{k-1} dP(x) = \mathbb{E}_x[C_x^{k-1}]. \tag{12}$$

Therefore, in order to predict the expected accuracy it suffices to estimate the values of $C_x$. A consequence of the result above is that the expressions that Kay et al. (2008) and Zheng et al. (2018) estimate (Equations 2 and 4 respectively) are in fact the same. Nevertheless, their estimation methods differ significantly.

Finally, we note that the theoretical connection between the reversed ROC and the evaluation of classification models extends beyond this particular work. For example, by plugging $k = 2$ into Theorem 1 it immediately follows that the area under the reversed ROC curve (rAUC) is the expected accuracy of two classes: $\text{rAUC} := \int_0^1 \overline{\text{rROC}}(u) \, du = \mathbb{E}_2[\mathcal{A}]$.

## 4 EXPECTED ACCURACY PREDICTION

In this section we present a new algorithm, *CleaneX*, for the prediction of the expected accuracy of a classifier. The algorithm is based on a neural network that estimates the values of $C_x$ using the classifier's scores on data from the available $k_1$ classes. These estimations are then used, based on the results of the previous section, to predict the classification accuracy at $k_2 > k_1$ classes.

The general task of estimating densities using neural networks has been widely addressed (Magdon-Ismail & Atiya, 1999; Papamakarios et al., 2017; Dinh et al., 2016; Uria et al., 2014). However, in our case, we need to estimate the cumulative distribution only at the value of the correct score $S_{y^*}(x)$. This is an easier task to perform and it allows us to design an estimation technique that learns to estimate the CDF in a supervised manner. We use a neural network $f(\cdot; \theta) : \mathbb{R}^{k_1} \to \mathbb{R}$ whose input is a vector of the correct score followed by the $k_1 - 1$ incorrect scores, and its output is $\hat{C}_x$. Once $\hat{C}_x$ values are estimated for each $x$, the expected accuracy for each $2 \leq k \leq k_1$ can be estimated according to Equation 12: $\hat{\mathbb{E}}_k[\mathcal{A}] = \frac{1}{N} \sum_x \hat{C}_x^{k-1}$, where $N = rk_1$. In each iteration, the network's weights $\theta$ are optimized by minimizing the average over $2 \leq k \leq k_1$ square distances from the observed accuracies $\bar{\mathcal{A}}_k^{k_1}$: $\frac{1}{k_1 - 1} \sum_{k=2}^{k_1} \left( \frac{1}{N} \sum_x \hat{C}_x^{k-1} - \bar{\mathcal{A}}_k^{k_1} \right)^2$. After the weights have converged, the estimated expected accuracy at $k_2 > k_1$ classes can be calculated by $\hat{\mathbb{E}}_{k_2}[\mathcal{A}] = \frac{1}{N} \sum_x \hat{C}_x^{k_2-1}$. Note that regardless of the target $k_2$, the network's input size is $k_1$. The proposed method is described in detail in Algorithm 1, below.

Unlike KDE and non-parametric regression, our method does not require a choice of basis functions or kernels. It does require choosing the network architecture, but as will be seen in the next sections, in all of our experiments we use the same architecture and hyperparameters, indicating that our algorithm requires very little tuning when applied to new classification problems.

Although the neural network allows more flexibility compared to the non-parametric regression, the key difference between the method we propose and previous ones is that *CleaneX* combines two sources of information: the classification scores (which are used as the network's inputs) and the average accuracies of the available classes (which are used in the minimization objective). Unlike the KDE and regression based methods which use only one type of information, we estimate the CDF of the incorrect scores evaluated at the correct score, $C_x$, but calibrate the estimations to produce accuracy curves that fit the observed ones. We do so by exploiting the mechanism of neural networks to iteratively optimize an objective function over the whole dataset.

---

**Algorithm 1:** *CleaneX*

---

**Input** : The classifier's score function $S$, a training set of $N$ examples $x$ from the set of $k_1$ available classes, target number of classes $k_2$; a feedforward neural network $f(\cdot; \theta)$, initial network weights $\theta_0$, number of training iterations $J$, learning rate $\eta$.

**Output:** Estimated accuracy at $k_2$ classes.

**for** $k = 2, \ldots, k_1$ **do**

  | Compute $\bar{\mathcal{A}}_k^{k_1}$

**end**

**for** *each $x$ in training set* **do**

  | Set $S'(x) \leftarrow \left( S_{y_1'}(x), \ldots, S_{y_{k_1-1}'}(x) \right)$

  | Sort $S'(x)$

  | Set $S_x \leftarrow (S_{y^*}(x), S'(x))$

**end**

**for** $j = 1, \ldots, J$ **do**

  **for** *each $x$* **do**

    | Set $\hat{C}_x \leftarrow f(S_x; \theta_j)$

  **end**

  Update network parameters performing a gradient descent step:

$$\theta_j \leftarrow \theta_{j-1} - \eta \nabla_\theta \left( \tfrac{1}{k_1-1} \sum_{k=2}^{k_1} \left( \tfrac{1}{N} \sum_x \hat{C}_x^{k-1} - \bar{\mathcal{A}}_k^{k_1} \right)^2 \right)$$

**end**

Return $\hat{\mathbb{E}}_{k_2}[\mathcal{A}] = \tfrac{1}{N} \sum_x \hat{C}_x^{k_2-1}$

---

## 5   SIMULATIONS AND EXPERIMENTS

Here we compare *CleaneX* to the KDE method (Kay et al., 2008) and to the non-parametric regression (Zheng et al., 2018) on simulated datasets that allow to examine under which conditions each method performs better, and on real-world datasets. Our goal is to predict the expected accuracy $\mathbb{E}_k[\mathcal{A}] \in [0, 1]$ for different values of $k$. Therefore, for each method we compute the estimated expected accuracies $\hat{\mathbb{E}}_k[\mathcal{A}]$ for $2 \leq k \leq k_2$ and measure the success of the prediction using the root of the mean squared error (RMSE): $\left( \frac{1}{k_2-1} \sum_{k=2}^{k_2} (\hat{\mathbb{E}}_k[\mathcal{A}] - \bar{\mathcal{A}}_k^{k_1})^2 \right)^{1/2}$.

For our method, we use in all the experiments an identical feed-forward neural network with two hidden layers of sizes 512 and 128, a rectified linear activation between the layers, and a sigmoid applied on the output. We train the network according to Algorithm 1 for $J = 10,000$ iterations with learning rate of $\eta = 10^{-4}$ using Adam optimizer (Kingma & Ba, 2014). For the regression based method we choose a radial basis and for the KDE based method a normal kernel, as recommended by the authors. The technical implementation details are provided in Appendix C.

A naive calculation of $\bar{\mathcal{A}}_k^K$ requires averaging the obtained accuracy over all possible subsets of $k$ classes from $K$ classes. However, Zheng et al. (2018) showed that for marginal classifiers $\bar{\mathcal{A}}_k^K = \frac{1}{\binom{K}{k}} \frac{1}{rk} \sum_x \binom{R_x}{k-1}$, where the sum is taken over all data points $x$ from the $K$ classes, and $R_x$ is the number of incorrect classes that the correct class of $x$ outperforms, that is $R_x = \sum_{y'} \mathbb{1}\{S_{y^*}(x) > S_{y'}(x)\}$. We use this result to compute $\bar{\mathcal{A}}_k^{k_2}$ values.

### 5.1   SIMULATION STUDIES

Here we provide comparison of the methods under different parametric settings. We simulate both classes and data points as $d$-dimensional vectors, with $d = 5$ (and $d = 3, 10$ shown in Appendix B). Settings vary in the distribution of classes $Y$ and data-points $X|Y$, and in the spread of data-points around the class centroids. We sample the classes $y_1, \ldots, y_{k_2}$ from a multivariate normal distribution

$\mathcal{N}(0, I)$ or a multivariate uniform distribution $\mathcal{U}(-\sqrt{3}, \sqrt{3})^3$. We then sample $r = 10$ data points for each class, either from a multivariate normal distribution $\mathcal{N}(y, \sigma^2 I)$ or from a multivariate uniform distribution $\mathcal{U}(y - \sqrt{3\sigma^2}, y + \sqrt{3\sigma^2})$. The difficulty level is determined by $\sigma^2 = 0.1, 0.2$. The classification scores $S_y(x)$ are set to be the euclidean distances between $x$ and $y$ (in this case the correct scores are expected to be lower than the incorrect ones, requiring some straightforward modifications to our analysis). For each classification problem, we subsample 50 times $k_1$ classes, for $k_1 = 100, 500$, and predict the accuracy at $2 \le k \le k_2 = 2000$ classes.

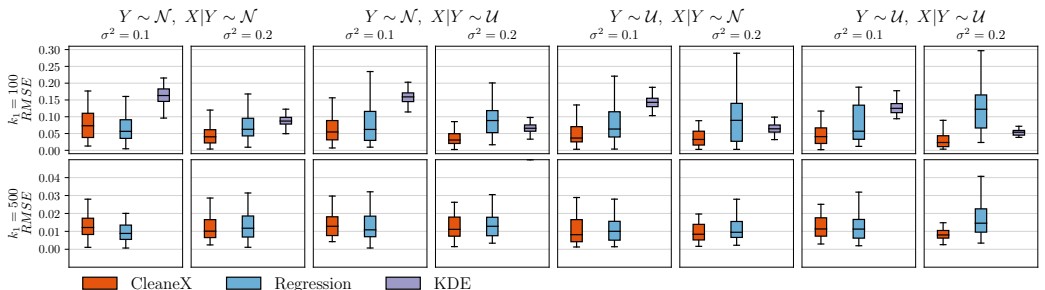

Figure 2: Simulation results. For each scenario we show a boxplot representing the RMSE values obtained over 50 repetitions using *CleaneX* (left box, orange), regression based method (middle box, blue) and KDE (right box, purple). The boxes extend from the lower to the upper quartile values, with a line at the median; whiskers show values at a distance of at most 1.5 IQR (interquartile range) from the lower and the upper quartiles; outliers are omitted from the figure for clarity. For $k_1 = 500$ the KDE based method always achieves RMSE values higher than 0.05 and is therefore not shown.

The results of the simulations, summarized in Figure 2, show the distribution of RMSE values for each method and setting over 50 repetitions. The corresponding accuracy curves are shown in Figure 4 in Appendix B. It can be seen that extrapolation to 20 times the original number of classes can be achieved with reasonable errors (the median RMSE is less than $5\%$ in almost all scenarios). Our method performs better or similar to the competing methods, often with substantial gain. For example, for $d = 5$ the results show that our method outperforms the KDE based method in all cases; it outperforms the regression based method in 7 out of 8 settings for $k_1 = 100$, and for $k_1 = 500$ the regression based method and *CleaneX* achieve very similar results. As can be seen in Figures 2 and 4, the accuracy curves produced by the KDE method are highly biased and therefore it consistently achieves the worst performance. The results of the regression method, on the other hand, are more variable than our method, especially at $k_1 = 100$. Additional results for $d = 3, 10$ (see Appendix B) are consistent with these results, though all methods predict better for $d = 10$.

## 5.2 EXPERIMENTS

Here we present the results of three experiments performed on datasets from the fields of computer vision and computational neuroscience. We repeat each experiment 50 times. In each repetition we sub-sample $k_1$ classes and predict the accuracy at $2 \le k \le k_2$ classes.

**Experiment 1 - Object Detection (CIFAR-100)** We use the CIFAR dataset (Krizhevsky et al., 2009) that consists of $32 \times 32$ color images from 100 classes, each class containing 600 images. Each image is embedded into a 512-dimensional space by a VGG-16 network (Simonyan & Zisserman, 2014), which was pre-trained on the ImageNet dataset (Deng et al., 2009). On the training set, the centroid of each class is calculated and the classification scores for each image in the test set are set to be the euclidean distances of the image embedding from the centroids. The classification accuracy is extrapolated from $k_1 = 10$ to $k_2 = 100$ classes.

**Experiment 2 - Face Recognition (LFW)** We use the "Labeled Faces in the Wild" dataset (Huang et al., 2007) and follow the procedure described in Zheng et al. (2018): we restrict the dataset to the 1672 individuals for whom it contains at least 2 face photos and include in our data exactly 2 randomly chosen photos for each person. We use one of them as a label $y$, and the other as a data

---

[3]The choice of $\sqrt{3}$ results in class covariances that equal to those in the multivariate normal case.

point $x$, consistent with a scenario of single-shot learning. Each photo is embedded into a 128-dimensional space using OpenFace embedding (Amos et al., 2016). The classification scores are set to be the euclidean distance between the embedding of each photo and the photos that are used as labels. Classification accuracy is extrapolated from $k_1 = 200$ to $k_2 = 1672$ classes.

**Experiment 3 - Brain Decoding (fMRI)** We analyze a "mind-reading" task described by Kay et al. (2008) in which a vector of $v = 1250$ neural responses is recorded while a subject watches $n = 1750$ natural images inside a functional MRI. The goal is to identify the correct image from the neural response vector. Similarly to the decoder in (Naselaris et al., 2011), we use $n_t = 750$ images and their response vectors to fit an embedding $g(\cdot)$ of images into the brain response space, and to estimate the residual covariance $\Sigma$. The remaining $n - n_t = k_2 = 1000$ examples are used as an evaluation set for $g(\cdot)$. For image $y$ and brain vector $x$, the score is the negative Mahalanobis distance $-\|g(y) - x\|_\Sigma^2$. For each response vector, the image with the highest score is selected. Classification accuracy is extrapolated from $k_1 = 200$ to $k_2 = 1000$ classes.

The experimental results, summarized in Figure 3.D, show that generally *CleaneX* outperforms the regression and KDE methods: Figure 3.E shows that the median ratio between RMSE values of the competing methods and CleaneX is higher than 1 in all cases except for KDE on the LFW dataset. As can be seen in Figure 3.A and B, the estimated accuracy curves produced by *CleaneX* have lower variance than those of the regression method (which do not always decrease with $k$), and on average our method outperforms the regression, achieving lower prediction errors by $18\%$ (CIFAR), $32\%$ (LFW) and $22\%$ (fMRI). On the other hand, the KDE method achieves lower variance but high bias in two of three experiments (Figure 3.C), where it is outperformed by *CleaneX* by $7\%$ (CIFAR) and $73\%$ (fMRI). However, in contrast to the simulation results and its performance on the CIFAR and fMRI datasets, the KDE method achieves exceptionally low bias on the LFW dataset, outperforming our method by $38\%$. Overall, *CleaneX* produces more reliable results across the experiments.

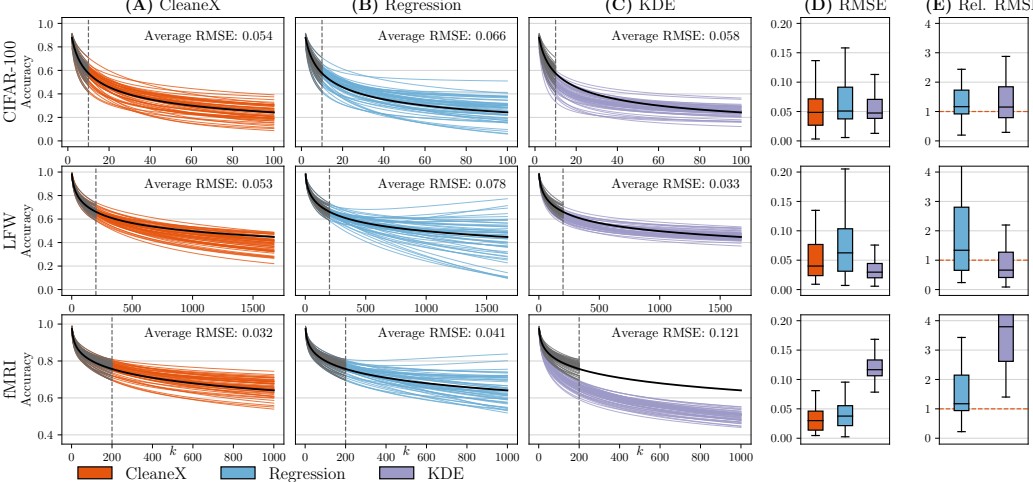

Figure 3: Experimental results. A, B, C: accuracy curves of the three datasets as predicted by *CleaneX*, regression and KDE, respectively; dotted vertical lines denote $k_1$, grey curves correspond to $\bar{\mathcal{A}}_k^{k_1}$ at each repetition, black curves correspond to $\mathbb{E}_k[\mathcal{A}]$ for $2 \le k \le k_2$; average RMSE is taken over all 50 repetitions. D: distribution of RMSE values over the 50 repetitions. E: distribution of the ratio between RMSE values of competing methods and *CleaneX* – values above 1 (orange dotted line) indicate that *CleaneX* outperforms the competing methods (charts capped at 4). In D and E, boxes show lower quartile, higher quartile and median; whiskers show values at 1.5 IQR from box; outliers are omitted from the figure for clarity.

## 6  DISCUSSION

In this work we presented the reversed ROC and showed its connection to the accuracy of marginal classifiers. We used this result to develop a new method for accuracy prediction.

Analysis of the two previous methods for accuracy extrapolation reveals that each of them uses only part of the relevant information for the task. The KDE method estimates $C_x$ based only on the scores, ignoring the observed accuracies of the classifier. Even when the estimates of $C_x$ are unbiased, the exponentiation in $\mathbb{E}_x[C_x^{k-1}]$ introduces bias. Since the estimation is not calibrated using the observed accuracies, the resulting estimations are often biased. As can be seen in Figure 2, the bias is higher for $k_1 = 500$ than for $k_1 = 100$, indicating that this is aggregated when $k_1$ is small. In addition, we found the method to be sensitive to monotone transformations of the scores, such as taking squared-root or logarithm. In contrast, the non-parametric regression based method uses pre-chosen basis functions to predict the accuracy curves and therefore has limited versatility to capture complicated functional forms. Since it ignores the values of the classification scores, the resulting predicted curves do not necessarily follow the form of $\mathbb{E}_x[C_x^{k-1}]$, introducing high variance estimations. As can be seen in Figure 2, the variance is higher for $k_1 = 100$ compared to $k_1 = 500$, indicating higher variance for small values of $k_1$. This can be expected since the number of basis functions that are used to approximate the discriminability function depends on $k_1$.

In *CleaneX*, we exploit the mechanism of neural networks in a manner that allows us to combine both sources of information with less restriction on the shape of the curve. Since the extrapolation follows the known exponential form using the estimated $C_x$ values it is characterized with low variance, and since the result is calibrated by optimizing an objective that depends on the observed accuracies, our method has low bias, and therefore consistently outperforms the previous methods.

Comparing the results for $k_1 = 500$ between different dimensions ($d = 5$ in Figure 2 and $d = 3, 10$ in Figure 6) it can be seen that the bias of the KDE based method and the variance of the regression based method are lower in higher data dimensions. Zheng & Benjamini (2016) show that if both $X$ and $Y$ are high dimensional vectors and their joint distribution factorizes, then the classification scores of Bayesian classifiers (which are marginal) are approximately Gausssian. The KDE and the regression based methods use Gaussian kernel and Gaussian basis functions respectively, and are perhaps more efficient in estimating approximately Gaussian score distributions. Apparently, scores for real data-sets behave closer to low-dimensional data, where the flexibility of *CleaneX* (partly due to being a non-parametric method) is an advantage.

A considerable source of noise in the estimation is the selection of the $k_1$ classes. The $\bar{\mathcal{A}}_k^{k_1}$ curves diverge from $\mathbb{E}_k[\mathcal{A}]$ as the number of classes increases, and therefore it is hard to recover if the initial $k_1$ classes deviated from the true accuracy. The effect of choosing different initial subsets can be seen by comparing the grey curves and the orange curves that continue them, for example in Figure 3. We leave the design of more informative sampling schemes for future work.

To conclude, we found the method we present to be considerably more stable than previous methods. Therefore, it is now possible for researchers to reliably estimate how well the classification system they are developing will perform using representations learned only on a subset of the target class set.

Although our work focuses on marginal classifiers, its importance extends beyond this class of algorithms. First, preliminary results show that our method yields good estimates even when applied to (shallow) non-marginal classifiers such as multi-logistic regression. Moreover, if the representation can adapt as classes are introduced, we expect accuracy to exceed that of a fixed representation. We can therefore use our algorithm to measure the degree of adaptation of the representation. Generalization of our method to non marginal classifiers is a prominent direction for future work.

### ACKNOWLEDGMENTS

We thank Etam Benger for many fruitful discussions, and Itamar Faran and Charles Zheng for commenting on the manuscript. YS is supported by the Israeli Council For Higher Education Data-Science fellowship and the CIDR center at the Hebrew University of Jerusalem.

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

## A  PROOF OF THEOREM 1

We begin by proving the following simple proposition:

**Proposition 1.** *Given $x$, let*

$$G_1 = \left\{ y \mid S_y(x) > \sup_t \left\{ P_{Y'}\left(S_{y'}(x) > t\right) \geq u\right\}\right\} \tag{13}$$

*and*

$$G_2 = \left\{ y \mid P_{Y'}\left(S_{y'}(x) > S_y(x)\right) < u\right\}, \tag{14}$$

*then $G_1 = G_2$.*

*Proof.* Let $G_1^c$, $G_2^c$ be the complement sets of $G_1, G_2$, respectively. Then

$$G_1^c = \left\{ y \mid S_y(x) \leq \sup_t \left\{ P_{Y'}\left(S_{y'}(x) > t\right) \geq u\right\}\right\} \tag{15}$$

and

$$G_2^c = \left\{ y \mid P_{Y'}\left(S_{y'}(x) > S_y(x)\right) \geq u\right\}. \tag{16}$$

Since $P_{Y'}\left(S_{y'}(x) > t\right)$ is not increasing in $t$ we have

$$y \in G_1^c \Leftrightarrow S_y(x) \leq \sup_t \left\{ P_{Y'}\left(S_{y'}(x) > t\right) \geq u\right\}$$
$$\Leftrightarrow P_{Y'}\left(S_{y'}(x) > S_y(x)\right) \geq u \tag{17}$$
$$\Leftrightarrow y \in G_2^c$$

and therefore $G_1 = G_2$. $\qquad\square$

We now prove Theorem 1, which was presented in Section 3:

*Proof.* We show that $1 - \overline{\mathrm{rROC}}\,(1-u) = D(u)$, and the rest follows immediately according to Theorem 3 of Zheng et al. (2018) (see Equation 4):

$$1 - \mathrm{rROC}_x(1-u) = 1 - \mathbb{1}\left[ S_{y^*}(x) > \sup_t \left\{ P_{Y'}(S_{y'}(x) > t) \geq 1-u\right\}\right] \quad \text{(Definition 1)}$$
$$= 1 - \mathbb{1}\left[ P_{Y'}(S_{y'}(x) > S_{y^*}(x)) < 1-u\right] \quad \text{(Proposition 1)}$$
$$= \mathbb{1}\left[ P_{Y'}(S_{y'}(x) \leq S_{y^*}(x)) \leq u\right]$$
$$= \mathbb{1}\left[ P_{Y'}(S_{y'}(x) < S_{y^*}(x)) \leq u\right], \tag{18}$$

where $\mathbb{1}[\cdot]$ denotes the indicator function, and the last equality follows from the assumption that $S_{y'}(x) \neq S_y(x)$ almost surely. From here,

$$1 - \overline{\mathrm{rROC}}\,(1-u) \tag{19}$$
$$= \mathbb{E}_X\left[ \mathbb{1}\left( P_{Y'}(S_{y'}(x) < S_{y^*}(x)) \leq u\right)\right]$$
$$= P_X\left( P_{Y'}(S_{y'}(x) < S_{y^*}(x)) \leq u\right)$$
$$= D(u)$$

as required. $\qquad\square$

## B  SIMULATIONS - ADDITIONAL RESULTS

Figures 4 and 5 show the accuracy curves for which we presented summarized results in Figure 2.

In Figure 6 we provide additional simulation results for data in dimensions $d = 3, 10$. We simulate eight $d$-dimensional datasets of $k_2 = 2000$ classes and extrapolate the accuracy from $k_1 = 500$ classes. As in Section 5.1, the datasets are combinations of two distributions of $Y$ and two of $X \mid Y$, in two different levels of classification difficulty. We sample the classes $y_1, \ldots, y_{k_2}$ from a multivariate normal distribution $\mathcal{N}(0, I)$ or a multivariate uniform distribution $\mathcal{U}(-1, 1)$. We sample $r = 10$ data points for each class from a multivariate normal distribution $N(y, \sigma^2 I)$ or from

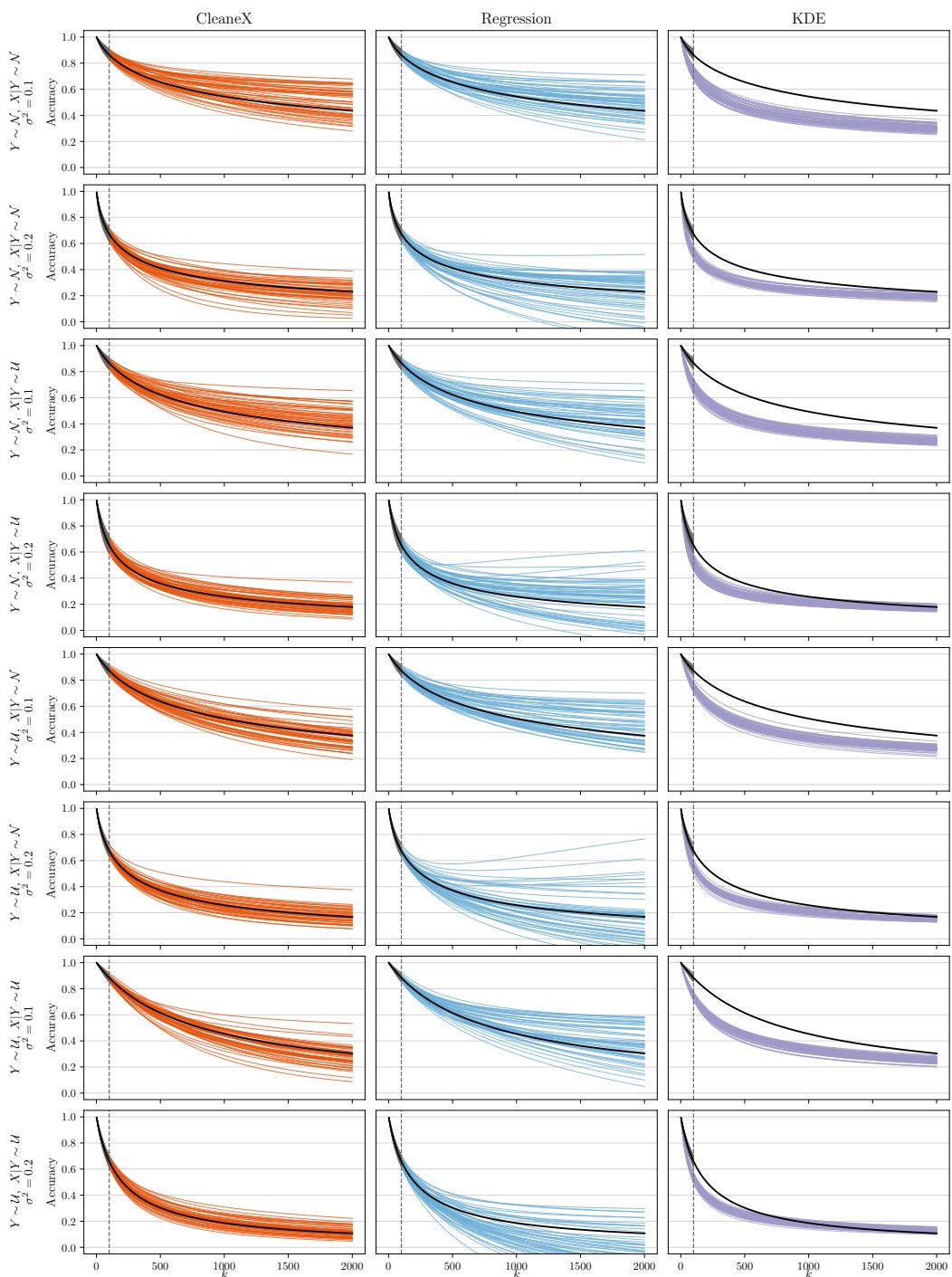

Figure 4: Comparison of predicted accuracy curves produced by *CleaneX* (left, orange), regression based method (middle, blue) and KDE (right, purple), on the eight simulated datasets with $d = 5$ and $k_1 = 100$ (dotted vertical line). The curves of $\bar{\mathcal{A}}_k^{k_1}$ for each repetition are shown in grey. The black curves correspond to $\mathbb{E}_k[\mathcal{A}]$ for $2 \le k \le k_2 = 2000$.

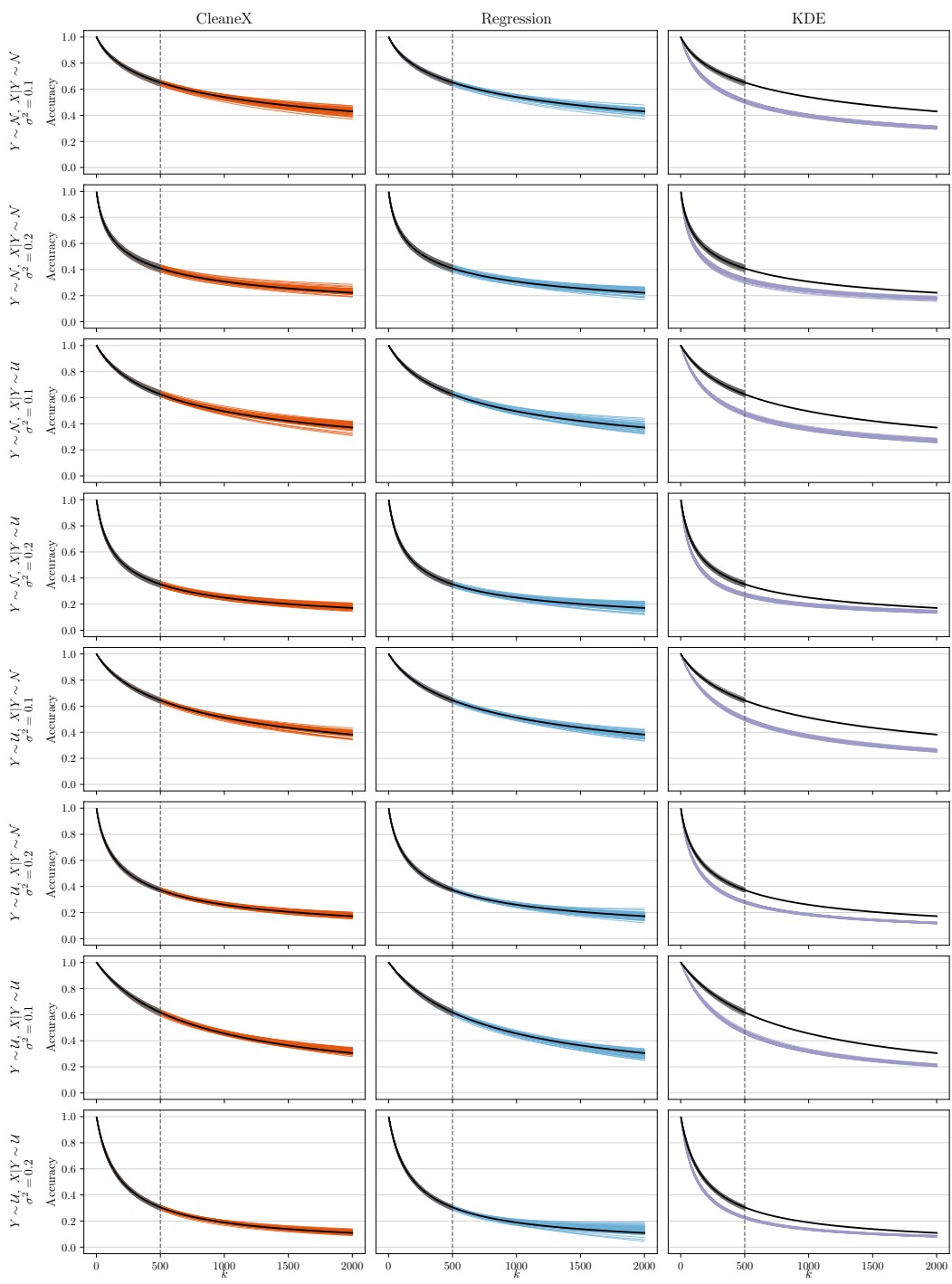

Figure 5: Comparison of predicted accuracy curves produced by *CleaneX* (left, orange), regression based method (middle, blue) and KDE (right, purple), on the eight simulated datasets with $d = 5$ and $k_1 = 500$ (dotted vertical line). The curves of $\bar{\mathcal{A}}_k^{k_1}$ for each repetition are shown in grey. The black curves correspond to $\mathbb{E}_k[\mathcal{A}]$ for $2 \le k \le k_2 = 2000$.

a multivariate uniform distribution $U(y - \sigma^2, y + \sigma^2)$. For $d = 3$ we set $\sigma^2$ to 0.1 for the easier classification task and 0.2 for the difficult one. For $d = 10$, the values of $\sigma^2$ are set to 0.6 and 0.9, respectively. The classification scores $S_y(x)$ are the euclidean distances between $x$ and $y$.

It can be seen that the results for $d = 3$ are consistent with those for $d = 5$. That is, our method outperforms the two competing methods, with an even bigger gap, in seven of the eight simulated datasets. For $d = 10$ the regression based method achieves comparable results to ours. As in the other simulations, our method produces less variance than the regression and less bias than KDE.

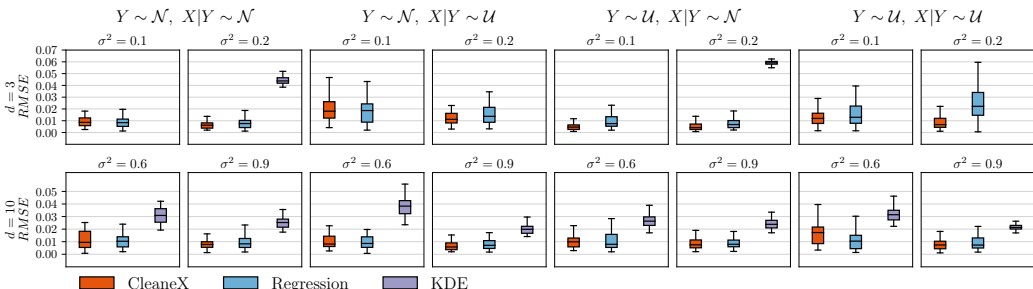

Figure 6: Additional simulation results. For each scenario we show a boxplot representing the RMSE values obtained over 50 repetitions using *CleaneX* (left box, orange), regression based method (middle box, blue) and KDE (right box, purple). The boxplots for the KDE method are not shown when all obtained RMSE values are higher than 0.07. The boxes extend from the lower to the upper quartile values, with a line at the median; whiskers show values at a distance of at most 1.5 IQR (interquartile range) from the lower and the upper quartiles; outliers are omitted from the figure for clarity.

## C  IMPLEMENTATION DETAILS

All the code in this work was implemented in Python 3.6. For the *CleaneX* algorithm we used TensorFlow 1.14; for the regression based method we used the `scipy.optimize` package with the "Newton-CG" method; kernel density estimation was implemented using the `density` function from the `stats` library in R, imported to Python through the `rpy2` package.

