# OpenReview forum: "Predicting Classification Accuracy When Adding New Unobserved Classes"
_ICLR.cc/2021/Conference — ICLR 2021 Poster_

### Official Review · AnonReviewer1 · 2020-10-19
**A promising novel solution**

**Rating:** 6
**Confidence:** 4

**Review:**

The authors discuss how a classifier’s performance over the initial class sample can be used to extrapolate its expected accuracy on a larger, unobserved set of classes by mean of the dual of the ROC function, swapping the roles of classes and samples. Grounded on such function, the authors develop a novel ANN approach learning to estimate the accuracy of classifiers on arbitrarily large sets of classes. Effectiveness of the approach is demonstrated on a suite of benchmark datasets, both synthetic and real-world.

The manuscript is well written and understandable also by a non-specialist audience; the reference list is up-to-date and the introduction properly details the motivations for tackling the problem. The underlying math is sound, and the proposed solution is smart, but the experimental section is not convincing, and hardly supporting the authors’ claims. Overall, I would vote for a weak accept.

Pros:
- The proposed solution is grounded and interesting, and the results shown are encouraging;
- When optimized/improved, CleaneX may have a relevant impact on the multi class classification theory,

Cons:
- Classifiers are compared on the basis of RMSE, although the original problem is multi class classification; I would strongly suggest a more classification-oriented measure such as multiclass MCC.
- CleaneX is compared only to regression and KDE - what about adding also very widespread algorithms such as RandomForest?
- Performance gain w.r.t. regression is quite limited, especially on real-world datasets: would CleaneX benefit from adding dropout layers or more refined activation functions?

---

> ### Author Response · Authors · 2020-11-12
> **Authors response to AnonReviewer1**
>
> We thank the reviewer for their time and constructive comments.
>
> > “The underlying math is sound, and the proposed solution is smart, but the experimental section is not convincing, and hardly supporting the authors’ claims”
>
> In almost all experiments and simulations, our method performs better or similar to the competing methods (on the real-world datasets it is outperformed only once by the KDE based method, and in the simulations it is slightly outperformed by the regression based method in the easier setting of two normal distributions). Unlike the other methods it has no catastrophic failures. We showed in the simulations section that the KDE method very often provides extremely biased results and the regression based method often is characterized with very high variance of predictions, making them unreliable.
> We realize that this was stated only briefly in the experimental section and will detail this in the updated version we will soon upload.
>
> Regarding con 1:
>
> Our paper discusses evaluating classifiers rather than fitting classifiers: we try to predict the classification accuracy that would be obtained if there were more classes. Therefore, we are not sure that we understand the comment on using multiclass MCC. Since our method tries to predict an accuracy level in the [0,1] range we measure success using RMSE. Predicting multiclass MCC for $k_2$ classes using $k_1$ classes instead of predicting balanced accuracy would require a non-trivial extension to this work.
>
> In our experiments, on each dataset we sample $k_1$ classes and pretend these are the available classes. We train a marginal classifier on those $k_1$ classes and using the classification scores it assigns, predict what would be the expected accuracy on sets of $k>k_1$ classes, for k between 2 and $k_2$. We measure the mean over k of the squared errors between the prediction and the true value. We repeat this 50 times and therefore report the average over the repetitions, hence the mean integrated square error.
> We will of course clarify that the results that we report are not for the multi-class classification error, but for the error between the predicted expected accuracy. However, if the reviewer recommends a different measure suitable for measuring the multiple prediction errors, we would certainly consider it.
>
> Regarding con 2:
>
> The algorithms (KDE, non-linear regression, NN) are not used for classification. Instead, they try to predict the expected accuracy of a given classifier by estimating a density function or a cumulative density function.  All three methods do not apply changes to the classification algorithm, but treat it as given.
> In our experiments, it seems that the neural-network is capable of perfectly fitting the given objective function. Random-forest does not naturally fit to density or c.d.f. estimation tasks, and we do not think there is much room for improvement without changing the objective function.
>
> Regarding con 3:
>
> In all three real-world data sets on which we performed experiments, our method outperforms the regression based method by achieving lower prediction errors. On average our error decreases by 11%, 18% and 31% using our method compared to the non-parametric regression method.
>
> We do not use the neural network as it is usually used, but as a mechanism to optimize an objective function over the whole dataset (in our setting there are no train and test sets). We optimize the estimations of $C_x$ so the estimated expected accuracies on $k$ classes fit the observed ones (for $k$ between 2 and $k_1$). For example, in the experiments presented in Figure 3, our goal is to fit each orange line to the corresponding grey line as closely as possible. It can be seen that our method already fits the lines almost perfectly, even when the same relatively simple network architecture is used across all the experiments. Of note is that using other activation functions or network parameters hardly affects the performance. We believe that the prediction error of our method can be attributed almost completely to the high variance between different samples of $k_1$ classes, and is therefore close to the best possible prediction in this setting. While it might seem that dropout can be used to reduce the effect of the high variance between the different class samples, our experiments showed it has no such effect.

---

### Official Review · AnonReviewer3 · 2020-10-29
**Okay, but lacking "content" and explanation**

**Rating:** 6
**Confidence:** 4

**Review:**

The authors show a relationship between classification accuracy and reverse ROC in multiclass classifiers, when there are new classes not seen in the training data. They propose a method called CleaneX that learns to estimate the accuracy of multiclass classifiers on arbitrarily large sets of classes.

Major Comments:

The concept seems rather novel; however, I am not very familiar with the literature in evaluating one-shot learning classifiers. The paper in general seems rather "empty" in that the authors simply show things, but do not explain the importance of them or spend time motivating the problem. For example, what practical use is there to know the expected accuracy of a classifier for predicting k classes? The use of knowing the accuracy for a specific class is obvious, but the overall accuracy is unclear.

The authors should not include the KDE in the box plots for Figure 2 for k_1=500 as they are much worse than the other two methods and stretches the axis making the other two methods very difficult to see. A note that the KDE performs much worse and isn't shown, or put in the appendix should suffice.

How is the black line representing the "true" accuracy curve calculated for the experiments?

For Figure 3, the authors should find a better way to display the performance that does not require eyeballing how well the colored lines follow the shape of the black one. Furthermore, it seems like for the LFW dataset, the KDE does much better overall, and for the CIFAR dataset both CleaneX and Regression seem to perform comparably.

In general, the authors provide some intuition on why the CleaneX method performs better than the others in the discussion, but this should be more closely tied to the experiments. E.g. the simulations should be pointed to, when the authors discuss how the accuracy curves converge to one-parameter families when dimensionality increases. There should also be more discussion on potentially why the CleaneX method is better than the others. E.g. why does the KDE method perform so poorly in high dimensionality; is it an artifact of being a non-parametric method in a "parametric" problem due to the simulations? Why does the Regression method have high variance occasionally?

UPDATE: Based on the Author's response, I have updated my score.

---

> ### Author Response · Authors · 2020-11-12
> **Authors response to AnonReviewer3**
>
> We thank the reviewer for the time and effort spent reviewing our paper, and for the detailed suggestions.
>
> 1. On motiviation: We will address this in an updated version. For multiclass classification, data acquisition and curation is often the most expensive resource. This is especially true for non-standard data, e.g in building brain-machine interfaces, shoe-print recognition systems, or even face-recognition software with a new camera. In these cases, practitioners may want to evaluate a classification system (data-collection apparatus, pre-learned representation) on a smaller number of classes to evaluate the feasibility of their approach and to compare different approaches.  Obviously, classification accuracy will deteriorate as the number of classes increases, but previous work shows that this deterioration is not “uniform” across classifiers and data-distributions. The goal of our method (and competing methods) is to use the pilot set of classes, say photos from $k_1 = 500$ people, to predict the expected accuracy over the choices of additional people, at deployment for the larger set of $k_2 = 5000$ people.
>
> 2. In the new version we will remove the KDE based method from the box plots and will explain that it is due to the fact that among all scenarios, the minimal RMSE value achieved by their method is 0.499 and therefore incomparable to the regression based method and to ours.
>
> 3. The black line represents at each point k the expected accuracies over all possible subsets of k classes. A naive calculation is computationally expensive, however for marginal classifiers, Zheng et al. showed that it can be calculated using the binomial coefficient of choosing k classes from $k_2$ and the rankings between the classification scores of each data point. We performed the calculation using this result and we will of course include the corresponding formula in the updated version.
>
> 4. We felt that including at least one figure presenting the accuracy curves in the main text will provide an important intuition regarding the manner in which each method performs. Specifically, how the high bias of the KDE method, and the high variance of the regression based method, are expressed in the predictions. However, we understand that it is hard to follow, therefore we will try to add another figure to facilitate the comparison, possibly a boxplot similar to the one we showed for the simulations. If the reviewer recommends a different  way to display the performance, we would certainly consider it.
>
> 5. Accuracy curves converge to a one-parameter family when dimensionality increases, due to central limit behaviour. [Zheng and Benjamini] show that if both $X$ and $E[X|Y]$ are high dimensional, scores for Bayes classifiers become approximately normal. The gaussian shape of the correct and incorrect-class scores should make prediction easier for all three methods. We will add an explanation regarding this in the updated version, and point to the relevant simulations.
>
> 6. The main reason that CleaneX is better than the other methods is because it combines their advantages by both using the density scores and the accuracies on the available classes, while each of the other methods uses only one of those (please see our answer to reviewer 4).
> Even if the estimator of $C_x$ is unbiased, $\hat{C}_x^{k-1}$ will incur large biases. However, in large values of k the true accuracy of the classifier changes significantly less when additional classes are added, providing the KDE based method an advantage.
> The regression method has two pitfalls. On the one hand, it has limited versatility to capture complicated functional forms because it uses a predetermined functional basis. On the other hand, It does not use much of the available information -- the actual correct and incorrect class scores -- and therefore may be unstable in some settings.
> In the new version of the text we will address all these questions, with the hope that the answers become cleaner and with more details.

---

### Official Review · AnonReviewer4 · 2020-10-29
**This paper introduces the reverse ROC concept and its role in predicting the accuracy of marginal classifiers on a large number of classes.**

**Rating:** 6
**Confidence:** 3

**Review:**

Quality:
Rigorous treatments of the ROC and reverse ROC concepts were offered. Cumulative distribution of class scores were used to estimate accuracy on a multi-class classification problem.

Clarity: The paper was well written and organized.

Originality/Significance:
The reverse ROC concept and its role in predicting accuracy of marginal classifiers for a large number of unknown classes seems to be original. Technical novelty is limited but the proposed approach could have a decent impact in the literature for a variety of problems involving large number of classes such as extreme classification, zero and one shot learning, open-world classification, multi-task learning etc.

Detailed Comments:
This paper shows that for marginal classifiers the expected accuracy can be estimated in terms of the CDF of incorrect scores, i.e., the probability of the correct class to outscore a randomly chosen incorrect one for a data point x, which is denoted by C_x. The proposed work trains a simple neural network to learn the mapping from class scores of x onto C_x. NN is trained to minimize the difference between the expected accuracies obtained from C_x and true accuracies predicted for the first k_1 classes. Once NN is trained it can be used to obtain C_x for k_2 classes (k_2>>k_1), which are in turn used to predict accuracy for the k_2 classes.

The paper introduces the reverse ROC concept to offer an interesting interpretation to expected accuracy of k classes. Unlike earlier work that uses KDE and non-parametric regression to predict the classification accuracy for a larger number of classes the proposed work uses a neural network for the same task.

Experiments performed on three different datasets (CIFAR100, Face recognition, and Brain decoding) suggest the proposed technique significantly outperforms KDE-based technique in predicting accuracy on a larger class set and also seems to be slightly more competitive than non-parametric regressio method.

The main technical novelty lie in the reverse ROC interpretation of the average accuracy. The NN part is just a substitute for the nonlinear regression Zheng et al. used. Both aims to achieve the same thing (learning a nonlinear mapping between class scores and CDF evaluation at the correct score) and results between the two techniques also seems to be very consistent.

In Algorithm 1 I am not quite sure how a neural net with a fixed number of input nodes deals with varying number of inputs? C_x is the CDF of incorrect scores evaluated at the correct score. The size of the input is k but k changes from 1 to k_1. When k is not equal to k1 there are k scores to be mapped to and therefore the input to the NN has to be k but k changes for every A_{k}.  Different subsets of the neural net weights have to be used to get \bar{C}_x^(k-1) for different k's, which does not make much sense. Need some clarification here.

---

> ### Author Response · Authors · 2020-11-12
> **Authors response to AnonReviewer4**
>
> We thank the reviewer for this thoughtful review.
>
> Comment 1:
>
> The reverse ROC and its interpretation of the average accuracy is indeed one of the two main contributions we present, and is the basis of the rest of our analysis. However, we would like to address your comment regarding the NN being a substitute for the nonlinear regression: the novelty of the NN based algorithm we present isn’t simply in the use of the NN instead of a regression, but in the manner in which it is constructed, that allows us to combine the advantages of the two previous methods as we explain below.
>
> The nonlinear regression (Zheng et al) tries to fit the k-class average accuracies curve; it optimizes the regression coefficients so that pre-chosen basis functions will fit those average accuracies. However, it ignores the information in the classification score values. In contrast, the method of Kay et al uses only the densities of the classification scores and estimates their parameters using KDE. Then for each x they integrate the estimated density of the incorrect scores up to the correct one, obtaining the probability of the correct class to outscore a randomly chosen incorrect one ($C_x$). Their method tries to best reconstruct the score densities and ignores the observed accuracies.
>
> Our NN method uses both sources of information: the correct and incorrect classification scores for each x are the inputs to the network; whereas the loss function depends on the observed accuracies at increasing k’s. Hence, we estimate the probabilities to outscore a random wrong class $C_x$, but calibrate this estimation to produce accuracy curves that fit the observed ones. We believe that combining these two sources is the main reason why our method consistently outperforms the other methods. (A second reason is that NN formulation gives more flexibility compared to the non-parametric regression; we can support this flexibility by using the extra information from the scores.)
> We will soon update the manuscript to better explain these points.
>
> Comment 2:
>
>  Regarding the number of inputs to the NN: the NN is always uses all $k_1$ available classes. The interpolation to $k<k_1$ or extrapolation to $k>k_1$ is through the $C_x$ values that have already been computed. Once the set of $C_x$ values is available, we can average $C_{x}^ {k-1}$ values to predict accuracy at $k$ classes.
>
> In more details, the input to the network in Algorithm 1 is a vector of  the correct score, and $k_1-1$ incorrect scores. The network’s output is a single scalar which is the estimation of $C_x$ for that particular $x$.
> To compute the loss for each $ 2 \leq k \leq k_1$ the following steps are performed:
>
> (a) predict $\hat{C}_x^{k-1}$ for each x of the $N$ training examples by a feedforward run of the
>       network
> (b) raise each $C_x$ to the $(k-1)$-th power: $\hat{C}^{k-1}$
> (c) compute the average of the powers: $\frac{1}{N} \sum_x \hat{C}^{k-1}$
> (d) compute the square distance between the average of the powers and the average accuracy  over all subsets of $k$ classes from the sample: $\bar{\mathcal{A}}^{k_1}_k$
>
> After the square distances are computed for each $k$, their sum over $k$ is minimized. In order to compute the predicted accuracy at some $k_2$ we use average of $C_x^ {k_2-1}$ values.
>
> We thank the reviewer for pointing this out and will clarify this in the updated version.

---

### Author Response · Authors · 2020-11-15
**Revised version uploaded**

Dear reviewers,

We thank you again for your efforts in reviewing this paper, and for the constructive comments.
We have uploaded a revised version, which now includes:

- Additional explanations regarding the motivation of our work and its contribution in the introduction section, due to a comment of AnonReviewer3.
- An additional explanation in Section 4 on why the NN is not simply a substitute to the regression, but is constructed in a manner that allows us to combine the advantages of the two previous methods, as well as an explanation regarding the input size of the NN, due to comments of AnonReviewer4.
- We merged the sections of the simulations and experiments. In the beginning of the merged section we added an explanation why RMSE is used to compare results (due to a comment of AnonReviewer1) and the formula according to which the black line is computed (due to a comment of AnonReviewer3).
- We rephrased the results to state more clearly how our method outperforms the previous methods, both on simulations and real-world datasets, due to comments of AnonReviewer1.
- We updated the boxplot figures (Figures 2 and 6) to exclude the KDE method when it isn’t comparable to the others, according to a comment of AnonReviewer3.
- We added two new sub-figures in Figure 3 to facilitate the comparison between the methods on real-world datasets, according to a comment of AnonReviewer3.
- We expanded the discussion section, which now includes a detailed comparison of the methods pointing to the simulation results, a better explanation regarding the one parameter family argument, and the reasons why our method outperforms the previous ones, due to comments of AnonReviewer3.

Please do not hesitate to post further comments or questions.

---

### Decision · Program_Chairs · 2021-01-07
**Final Decision**

**Decision:**

Accept (Poster)

**Comment:**

The paper is very clear.  It provides a good overview of the problem, making it easy to follow even for researchers outside the area.

This work provides a novel approach for extrapolating the expected accuracy on a larger set of classes from a training set with smaller number of classes with a creative, simple and elegant solution through reversed ROC.  Such an approach will be useful for extreme classification settings.  In real-world settings, classifiers are often trained on a pilot set of data, and then deployed where the classes are much larger.  It is useful to have a mechanism to estimate how the classification performance will change with larger number of classes.

The reviewers all agree that this work provides a novel contribution to predicting classification accuracy.  The authors have satisfactorily addressed the reviewers’ comments and provided sufficient clarification to the questions.  We also appreciate the edits that the authors have made.